# Variance-Covariance Regularization Improves Representation Learning

## Abstract

Transfer learning plays a key role in advancing machine learning models, yet conventional supervised pretraining often undermines feature transferability by prioritizing features that minimize the pretraining loss. Recent progress in self-supervised learning (SSL) has introduced regularization techniques that bolster feature transferability. In this work, we adapt an SSL regularization technique from the VICReg method to supervised learning contexts, introducing Variance-Covariance Regularization (VCReg). This adaptation encourages the network to learn a high-variance, low-covariance representation, promoting the learning of more diverse features. We outline best practices for implementing this regularization framework into various neural network architectures and present an optimized strategy for regularizing intermediate representations. Through extensive empirical evaluation, we demonstrate that our method significantly enhances transfer learning, achieving excellent performance across numerous tasks and datasets. VCReg also improves performance in scenarios like long-tail learning, and hierarchical classification. Additionally, we conduct analyses to suggest that its effectiveness may stem from its success in addressing challenges like gradient starvation and neural collapse. In summary, VCReg offers a universally applicable regularization framework that significantly advances the state of transfer learning, highlights the connection between gradient starvation, neural collapse, and feature transferability, and potentially opens a new avenue for regularization in this domain.

## 1 Introduction

Transfer learning enables models to apply knowledge from one domain to enhance performance in another, particularly when data are scarce or costly to obtain (Pan & Yang, 2010; Weiss et al., 2016; Zhuang et al., 2020; Bommasani et al., 2021). One of the key challenges arises during the supervised pretraining phase. In this phase, models often lack detailed information about the downstream tasks to which they will be applied. Nevertheless, they must aim to capture a broad spectrum of features beneficial across various applications (Bengio, 2012; Caruana, 1997; Yosinski et al., 2014). Without proper regularization techniques, these supervised pretrained models tend to overly focus on features that minimize supervised loss, resulting in limited generalization capabilities and issues such as gradient starvation and neural collapse (Zhang et al., 2016; Neyshabur et al., 2017; Zhang et al., 2021; Pezeshki et al., 2021; Papyan et al., 2020; Shwartz-Ziv, 2022).

To tackle these challenges we adapt the regularization techniques of the self-supervised VICReg method (Bardes et al., 2021) for the supervised learning paradigm. Our method, termed Variance-Covariance Regularization (VCReg), aims to encourage the learning of representations with high variance and low covariance, thus avoiding the overemphasis on features that merely minimize supervised loss. Crucially, our detailed tests reveal that the effectiveness of VCReg strongly depends on how well it is integrated into different neural network designs. Instead of simply applying VCReg to the final representation of the network, we explore the most effective ways to incorporate it throughout the intermediate representations of the network.

The structure of the paper is as follows: We begin with an introduction of our method, including an outline of a fast implementation strategy designed to minimize computational overhead. Following this, we present a series of experiments aimed at validating the method's efficacy across a wide

range of tasks, datasets, and neural network architectures. Subsequently, we conduct analyses on the learned representations to demonstrate VCReg's effectiveness in mitigating common issues in transfer learning, such as neural collapse and gradient starvation. This finding suggests a promising avenue for future research in transfer learning: focusing on resolving issues like gradient starvation and neural collapse, particularly in the context of transfer learning, has the potential to significantly improve performance.

Our paper makes the following contributions:

1. We introduce a robust strategy for applying VCReg to neural networks, including integrating it into the intermediate layers.

2. We propose a computationally efficient implementation of VCReg. This implementation is optimized to ensure minimal impact from additional computational overhead, allowing for seamless integration into existing workflows while maintaining high training speed and resource efficiency.

3. Through extensive experiments on benchmark datasets, we demonstrate that using VCReg yields notable improvements in transfer learning performance across various network architectures, including ResNet (He et al., 2016), ConvNeXt (Liu et al., 2022), and ViT (Dosovitskiy et al., 2020). Moreover, with preliminary results, we also find that VCReg could improve performance in scenarios like long-tail learning, and hierarchical classification.

4. We investigate the learned representation of VCReg, revealing its effectiveness in combating challenges such as gradient starvation (Pezeshki et al., 2021), neural collapse (Papyan et al., 2020), and information compression (Shwartz-Ziv, 2022).

## 2 RELATED WORK

### 2.1 VARIANCE-INVARIANCE-COVARIANCE REGULARIZATION (VICREG)

VICReg (Bardes et al., 2021) is a novel SSL method. VICReg encourages learned representations to be invariant to data augmentation. However, by optimizing only the invariant criterion, the network will learn to generate a constant representation for all inputs. This means the representations will be invariant not only to data augmentation, but also to the input itself.

VICReg primarily regularizes the network by using a combination of variance loss and covariance loss. The variance loss encourages high variance in the learned representations, thereby promoting the learning of a wide range of features. The covariance loss, on the other hand, aims to minimize redundancy in the learned features by reducing the overlap in information captured by different dimensions of the representation. This dual-objective optimization framework has been found to be effective in promoting diverse feature learning (Shwartz-Ziv et al., 2022). In this work, we borrow the feature collapse prevention mechanism from VICReg and propose the variance-covariance regularization method for supervised network training to improve transfer learning performance.

To calculate the loss function of VICReg with a batch of data $\{x_1, \ldots, x_n\}$, we first need to have a pair of inputs $(x_i', x_i'')$ such that $x_i'$ and $x_i''$ are two augmented versions of the original input $x_i$. With the neural network $f_\theta(\cdot)$ and the final representations $z_i' = f_\theta(x_i')$ and $z_i'' = f_\theta(x_i'')$, the VICReg minimizes the following loss (we defer the detailed formulation of the variance and covariance loss terms to the subsequent section where we introduce our methods):

$$\begin{aligned}
\ell_{\text{VICReg}}(z_1', \ldots, z_n', z_1'', \ldots, z_n'') = {} & \alpha \ell_{\text{var}}(z_1', \ldots, z_n') + \alpha \ell_{\text{var}}(z_1'', \ldots, z_n'') \\
& + \beta \ell_{\text{cov}}(z_1', \ldots, z_n') + \beta \ell_{\text{cov}}(z_1'', \ldots, z_n'') \\
& + \sum_{i=1}^{n} \ell_{\text{inv}}(z_i', z_i'')
\end{aligned} \tag{1}$$

Notice that the only loss term that requires two augmented images is the invariance loss. We usually avoid using two augmented images for each training step in supervised learning. This is because it would approximately double the total computation, as we would need to perform two forward passes at each step. Furthermore, as discussed in some previous works (Shwartz-Ziv, 2022; Shwartz-Ziv

et al., 2023), the invariance term is not the essential factor that helps diversify the features. Therefore, in our adaptation to the supervised regime, we omit the invariance term from the regularization.

## 2.2 REPRESENTATION WHITENING AND FEATURE DIVERSITY REGULARIZERS

Representation whitening is a technique for processing inputs before they enter a network layer. It transforms the input so that its components are uncorrelated with unit variance (Kessy et al., 2018). This transformation achieves enhanced model optimization and generalization. It uses a whitening matrix derived from the data's covariance matrix and results in an identity covariance matrix, thereby aiding gradient flow during training and acting as a lightweight regularizer to reduce overfitting and encourage robust data representations (LeCun et al., 2002).

In addition to whitening as a processing step, additional regularization terms can be introduced to enforce decorrelation in the representations. Various prior works have explored these feature diversity regularization techniques to enhance neural network training (Cogswell et al., 2015; Ayinde et al., 2019; Laakom et al., 2023). These methods encourage diverse features in the representation by adding a regularization term. Recent methods like WLD-Reg (Laakom et al., 2023) and De-Cov (Cogswell et al., 2015) also employ covariance-matrix-based regularization to promote feature diversity, similar to our approach.

However, the studies cited above primarily concentrate on the benefits of optimization and generalization for the source task, frequently overlooking their implications for transfer learning. VCReg sets itself apart by explicitly targeting enhancements in transfer learning performance. Our results indicate that such regularization techniques yield only modest performance improvements in in-domain evaluations. This may be attributed to the fact that modern optimizers and regularizers have already significantly alleviated challenges related to in-domain optimization and generalization. Therefore, the most impactful domain for these types of regularization appears to be transfer learning.

## 2.3 GRADIENT STARVATION AND NEURAL COLLAPSE

Gradient starvation and neural collapse are two recently recognized phenomena that can significantly affect the quality of learned representations and the network's generalization ability (Pezeshki et al., 2021; Papyan et al., 2020; Ben-Shaul et al., 2023). Gradient starvation occurs when certain parameters in a deep learning model receive very little gradient during the training process, thereby leading to slower or non-existent learning for these parameters (Pezeshki et al., 2021). Neural collapse, on the other hand, is a phenomenon observed during the late stages of training where the internal representations of the network tend to collapse towards each other, resulting in a loss of feature diversity (Papyan et al., 2020). Both phenomena are particularly relevant in the context of transfer learning, where models are initially trained on a source task before being fine-tuned for a target task. Our work, through the use of VCReg, seeks to mitigate these issues, offering a pathway to more effective transfer learning.

## 3 VARIANCE-COVARIANCE REGULARIZATION

### 3.1 VANILLA VCREG: AN INTRODUCTION TO THE BASIC FORMULATION

Consider a labeled dataset comprising $N$ samples, denoted as $\{(x_1, y_1), \ldots, (x_N, y_N)\}$ and a neural network $f_\theta(\cdot)$, which takes these inputs $x_i$ and produces final predictions $\tilde{y}_i = f_\theta(x_i)$. In standard supervised learning, the loss is defined as $L_{\text{sup}} = \frac{1}{N} \sum_{i=1}^{N} \ell_{\text{sup}}(\tilde{y}_i, y_i)$.

The core objective of Vanilla VCReg is to ensure that the $D$-dimensional input representation $h_i$ to the last layer of the network exhibit both high variance and low covariance. To achieve this, we employ variance and covariance losses, same as mentioned in equation 1:

$$\ell_{\text{vcreg}}(h_1, \ldots, h_N) = \alpha \ell_{\text{var}}(h_1, \ldots, h_N) + \beta \ell_{\text{cov}}(h_1, \ldots, h_N) \tag{2}$$

The variance and covariance loss functions are defined as:

$$\ell_{\text{var}} = \frac{1}{D} \sum_{i=1}^{D} \max(0, 1 - \sqrt{C_{ii}}) \tag{3}$$

$$\ell_{\text{cov}} = \frac{1}{D(D-1)} \sum_{i \neq j} C_{ij}^2 \tag{4}$$

where $C = \frac{1}{N-1} \sum_{i=1}^{N} (h_i - \bar{h})(h_i - \bar{h})^T$ denotes the covariance matrix, and $\bar{h}$ represents the mean vector, given by $\bar{h} = \frac{1}{N} \sum_{i=1}^{N} h_i$.

Intuitively speaking, the covariance matrix captures the interdependencies among the dimensions of the feature vectors $z_i$. Maximizing $\ell_{\text{var}}$ encourages each feature dimension to contain unique, non-redundant information, while minimizing $\ell_{\text{cov}}$ aims to reduce the correlation between different dimensions, thus promoting feature independence. The overall training loss then becomes:

$$L_{\text{vanilla}} = \alpha \ell_{\text{var}}(z_1, \ldots, z_N) + \beta \ell_{\text{cov}}(z_1, \ldots, z_N) + \frac{1}{N} \sum_{i=1}^{N} \ell_{\text{sup}}(\tilde{y}_i, y_i) \tag{5}$$

Here, $\alpha$ and $\beta$ serve as hyperparameters to control the strength of each regularization term.

## 3.2 EXTENDING VCREG TO INTERMEDIATE REPRESENTATIONS

While regularizing the final layer in a neural network offers certain benefits, extending this approach to intermediate layers via VCReg provides additional advantages. (For empirical evidence supporting this claim, please refer to Appendix A). Regularizing intermediate layers enables the model to capture more complex, higher-level abstractions. This strategy minimizes internal covariate shifts across layers, which in turn improves both the stability of training and the model's generalization capabilities. Furthermore, it fosters the development of feature hierarchies and enriches the latent space, leading to enhanced model interpretability and improved transfer learning performance.

To implement this extension, VCReg is applied at $M$ strategically chosen layers throughout the neural network. For each intermediate layer $j$, we denote the feature representation for an input $x_i$ as $h_i^{(j)} \in \mathbb{R}^{D_j}$. This culminates in a composite loss function, expressed as follows:

$$L_{\text{VCReg}} = \sum_{j=1}^{M} \left[ \alpha \ell_{\text{var}}(h_1^{(j)}, \ldots, h_N^{(j)}) + \beta \ell_{\text{cov}}(h_1^{(j)}, \ldots, h_N^{(j)}) \right] + \frac{1}{N} \sum_{i=1}^{N} \ell_{\text{sup}}(\tilde{y}_i, y_i) \tag{6}$$

**Spatial Dimensions** However, applying VCReg to intermediate layers of real-world neural networks presents challenges due to the spatial dimensions in these intermediate representations. Naively reshaping these representations into long vectors would lead to unmanageably large covariance matrices, thereby increasing computational costs and risking numerical instability. To address this issue, we adapt VCReg to accommodate networks with spatial dimensions. Each vector at a different spatial location is treated as an individual sample when calculating the covariance matrix. Both the variance loss and the covariance loss are then calculated based on this modified covariance matrix.

In terms of practical implementation, a VCReg is usually applied subsequent to each block within the neural network architecture, often succeeding residual connections. This placement allows for seamless incorporation into current network architecture and training paradigms.

**Addressing Outliers with Smooth L1 Loss** After treating spatial locations as independent samples for covariance computation, the resulting samples are no longer statistically independent. This can lead to outliers in the covariance matrix and unstable gradient updates. To address this, we introduce a smooth L1 penalty into the covariance loss term. Specifically, we replace the traditional squared covariance values $C_{ij}$ in $\ell_{\text{cov}}$ with a smooth L1 function:

$$\text{SmoothL1}(x) = \begin{cases} x^2, & \text{if } |x| \leq \delta \\ 2\delta|x| - \delta^2, & \text{otherwise} \end{cases} \tag{7}$$

By implementing this modification, we ensure that the loss function increases in a more controlled manner with respect to large covariance values. Empirically, this minimizes the impact of outliers, thereby enhancing the stability of the training process.

### 3.3 Fast Implementation

To optimize implementation speed, we take advantage of the fact that VCReg only affects the loss function and not the forward pass. This allows us to focus on directly modifying the backward function for improvements. Specifically, we sidestep the usual process of calculating the VCReg loss and subsequent backpropagation. Instead, we directly adjust the computed gradients, which is feasible since the VCReg loss calculation relies solely on the current representation. Further details of this speed-optimized technique are outlined in Appendix B.

We quantify the computational overhead by measuring the average time required for one NVIDIA A100 GPU to execute both the forward and backward passes on the entire network for a batch size of 128 using the ImageNet dataset. These results are summarized in Table 1. For the sake of comparison, we also include the latencies associated with adding Batch Normalization (BN) layers, revealing that our optimized VCReg implementation exhibits similar latencies to BN layers.

**Table 1: Average Time Required for One Forward and Backward Pass with Various Layers Inserted** Comparison of computational latencies across different configurations of ViT and ConvNeXt networks. The table demonstrates the efficacy of the optimized VCReg layer in terms of computational time, compared to both naive VCReg and Batch Normalization (BN) layers.

| Network | Number of Inserted Layers | Identity | VCReg (Naive) | VCReg (Fast) | BN |
|---|---|---|---|---|---|
| ViT-Base-32 | 12 | 0.223s | 1.427s | 0.245s | 0.247s |
| ConvNeXt-T | 18 | 0.442s | 2.951s | 0.471s | 0.468s |

## 4 Experiments

In this section, we initially outline the experimental framework and findings to highlight the effectiveness of our proposed regularization approach, VCReg, within the realm of transfer learning that utilizes supervised pretraining. Subsequent to that discussion, we extend our experiments beyond the scope of supervised pretraining to suggest that VCReg could be applicable across various learning paradigms. For guidelines on reproducing these experiments, please consult Appendix C.

### 4.1 Transfer Learning with Supervised Pretraining

In this section, we adhere to evaluation protocols established by seminal works such as (Chen et al., 2020; Kornblith et al., 2021; Misra & Maaten, 2020) for our transfer learning experiments.

Initially, we pretrained models using three different architectures: ResNet-50 (He et al., 2016), ConvNeXt-Tiny (Liu et al., 2022), and ViT-Base-32 (Dosovitskiy et al., 2020), on the full ImageNet dataset. We followed the standard PyTorch recipes (Paszke et al., 2019) for all networks and did not modify any hyperparameters other than those related to VCReg to ensure a fair baseline comparison. Subsequently, we performed a linear probing evaluation across a variety of datasets to evaluate the transfer learning performance.

For ResNet-50, we included two other feature diversity regularizer methods, namely DeCov (Cogswell et al., 2015) and WLD-Reg (Laakom et al., 2023), for comparison. We conducted experiments solely with ResNet-50 because it is the principal architecture used in the WLD-Reg paper. To ensure a fair comparison, we sourced hyperparameters from Laakom et al. (2023) for both DeCov and WLD-Reg.

The results presented in Table 2 depict significant improvements in transfer learning performance across all downstream datasets when VCReg is applied to the three architectures used. There is strong evidence to suggest that VCReg can help boost overall transfer learning performance, and it is effective for both ConvNet and Transformer architectures.

### 4.2 Beyond Transfer Learning with Supervised Learning

In this section, we explore the versatility of the VCReg regularization method by extending its application beyond transfer learning with supervised pretraining. We focus on three specialized

**Table 2: Transfer Learning Performance with ImageNet Supervised Pretraining** The table shows performance metrics for different architectures. Each model is pretrained on the full ImageNet dataset and then tested on different downstream datasets using linear probing. Application of VCReg consistently improves performance and beats other feature diversity regularizer.

| Architecture | ImageNet | iNat18 | Places | Food | Cars | Aircraft | Pets | Flowers | DTD |
|---|---|---|---|---|---|---|---|---|---|
| ResNet-50 | 76.1% | 42.8% | 50.6% | 69.1% | 43.6% | 54.8% | 91.9% | 77.1% | 68.7% |
| ResNet-50 (DeCov) | 75.9% | 43.1% | 50.4% | 69.0% | 45.7% | 55.5% | 90.6% | 79.2% | 69.1% |
| ResNet-50 (WLD-Reg) | **76.5%** | 43.9% | **51.2%** | 70.2% | 43.9% | 58.7% | 91.4% | 80.7% | 69.0% |
| ResNet-50 (VCReg) | 76.3% | **45.3%** | **51.2%** | **71.7%** | **54.1%** | **70.5%** | **92.1%** | **88.0%** | **70.8%** |
| ConvNeXt-T | **82.5%** | 51.6% | 53.8% | 78.4% | 62.9% | 74.7% | 93.9% | 91.3% | 72.9% |
| ConvNeXt-T (VCReg) | 82.4% | **52.3%** | **54.7%** | **79.6%** | **64.2%** | **76.3%** | **94.1%** | **92.7%** | **73.3%** |
| ViT-Base-32 | 75.9% | 39.1% | 47.9% | 70.6% | 51.2% | 63.8% | 90.3% | 84.6% | 66.1% |
| ViT-Base-32 (VCReg) | **76.3%** | **40.6%** | **48.1%** | **70.9%** | **52.0%** | **65.8%** | **91.0%** | **86.6%** | **66.5%** |

learning scenarios: 1) class imbalance via long-tail learning, 2) synergizing with self-supervised learning frameworks, and 3) hierarchical classification problems. The objective is to assess the adaptability of VCReg across various data distributions and learning paradigms, thereby evaluating its broader utility in machine learning applications.

**Class Imbalance with Long-Tail Learning** Class imbalance is a pervasive issue in many real-world datasets and poses a considerable challenge to standard neural network training algorithms. We conducted experiments to assess how well VCReg addresses this issue through long-tail learning. We evaluated VCReg using the CIFAR10-LT and CIFAR100-LT Krizhevsky et al. (2009) datasets, both engineered to have an imbalance ratio of 100. These experiments were conducted using a ResNet-32 backbone architecture. The per-class sample sizes ranged from 5,000 to 50 for CIFAR10-LT and from 500 to 5 for CIFAR100-LT.

**Table 3: Performance Comparison on Class-Imbalanced Datasets Using VCReg**: This table shows the accuracy of standard ResNet-32 with and without VCReg when trained on class-imbalanced CIFAR10-LT and CIFAR100-LT datasets. The VCReg-enhanced models show improved performance, demonstrating the method's effectiveness in addressing class imbalance.

| Training Methods | CIFAR10-LT | CIFAR100-LT |
|---|---|---|
| ResNet-32 | 69.6% | 37.4% |
| ResNet-32 (VCReg) | **71.2%** | **40.4%** |

Table 3 shows that models augmented with VCReg consistently outperformed the standard ResNet-32 models on imbalanced datasets. These results are noteworthy because they demonstrate that VCReg effectively enhances the model's ability to discriminate between classes in imbalanced settings. This establishes VCReg as a valuable tool for real-world applications where class imbalance is often a concern.

**Enhancing Self-Supervised Learning with VCReg** Our subsequent investigation focuses on examining the synergy between VCReg and existing self-supervised learning paradigms. We employed a ResNet-50 architecture, training it for 100 epochs under four different configurations: using either SimCLR loss or VICReg loss, coupled with the ImageNet dataset. For evaluation, we conducted linear probing tests on multiple downstream task datasets, following the protocols prescribed by Misra & Maaten (2020); Zbontar et al. (2021).

As indicated in Table 4, integrating VCReg into self-supervised learning paradigms such as SimCLR and VICReg resulted in consistent performance improvements for transfer learning. Specifically, the linear probing accuracies were enhanced across nearly all the evaluated datasets. These gains underscore the broad applicability and versatility of VCReg, demonstrating its potential to enhance various machine learning methodologies.

**Investigating Hierarchical Classification Capabilities** To evaluate the efficacy of the learned representations across multiple levels of class granularity, we conducted experiments on the CIFAR100 dataset as well as five distinct subsets of ImageNet (Engstrom et al., 2019). In each dataset, every data sample is tagged with both superclass and subclass labels, denoted as $(x_i, y_i^{\mathrm{sup}}, y_i^{\mathrm{sub}})$. Note

**Table 4: Impact of VCReg on Self-Supervised Learning Methods**: This table presents a comparative analysis of ResNet-50 models pretrained with SimCLR and VICReg losses on ImageNet, both with and without the VCReg applied. The models are evaluated using linear probing on various downstream task datasets. The VCReg models consistently outperform the non-VCReg models, showcasing the method's broad utility in transfer learning for self-supervised learning scenarios.

| Pretraining Methods | ImageNet | iNat18 | Places | Food | Cars | Aircraft | Pets | Flowers | DTD |
|---|---|---|---|---|---|---|---|---|---|
| SimCLR | 67.2% | 37.2% | 52.1% | 66.4% | 35.7% | **62.3%** | 76.3% | 82.6% | 68.1% |
| SimCLR (VCReg) | **67.1%** | **41.3%** | **52.3%** | **67.7%** | **40.6%** | 61.9% | **76.6%** | **83.6%** | **69.0%** |
| VICReg | 65.2% | **41.7%** | 48.2% | 61.0% | 27.3% | 51.2% | 79.1% | 74.3% | 65.4% |
| VICReg (VCReg) | **66.3%** | 41.4% | **49.6%** | **61.6%** | **29.3%** | **54.2%** | **79.7%** | **74.5%** | **66.5%** |

that while samples sharing the same subclass label also share the same superclass label, the reverse does not necessarily hold true. Initially, the model was trained using only the superclass labels, i.e., the $(x_i, y_i^{\text{sup}})$ pairs. Subsequently, linear probing was employed with the subclass labels $(x_i, y_i^{\text{sub}})$ to assess the quality of features abstracted at the superclass level.

**Table 5: Impact of VCReg on Hierarchical Classification in ConvNeXt Models:** This table summarizes the classification accuracies obtained with ConvNeXt models, both with and without the VCReg regularization, across multiple datasets featuring hierarchical class structures. The models were initially trained using superclass labels and subsequently probed using subclass labels. VCReg consistently boosts performance in subclass classification tasks.

| | CIFAR100 | Subsets of ImageNet | | | | |
|---|---|---|---|---|---|---|
| | | living_9 | mixed_10 | mixed_13 | geirhos_16 | big_12 |
| Superclass Count | 20 | 9 | 10 | 13 | 16 | 12 |
| Subclass Count | 100 | 72 | 60 | 78 | 32 | 240 |
| ConvNeXt | 60.7% | 53.4% | 60.3% | 61.1% | 60.5% | 51.8% |
| ConvNeXt (VCReg) | **72.9%** | **62.2%** | **67.7%** | **66.0%** | **70.1%** | **61.5%** |

Table 5 presents key performance metrics, highlighting the substantive improvements VCReg brings to subclass classification. The improvements are consistent across all datasets, with the CIFAR100 dataset showing the most significant gain—an increase in accuracy from 60.7% to 72.9%. These results underscore VCReg's capability to assist neural networks in generating feature representations that are not only discriminative at the superclass level but are also well-suited for subclass distinctions. This attribute is particularly advantageous in real-world applications where class categorizations often exist within a hierarchical framework.

# 5 Exploring the Benefits of VCReg

This section aims to thoroughly unpack the multi-faceted benefits of VCReg in the context of supervised neural network training. Specifically, we discuss its capability to address challenges such as gradient starvation (Pezeshki et al., 2021), neural collapse (Papyan et al., 2020), and the preservation of information richness during model training (Shwartz-Ziv, 2022).

## 5.1 Mitigating Gradient Starvation

In line with the original study on gradient starvation (Pezeshki et al., 2021), we observe that most traditional regularization techniques fall short of capturing the vital features for the 'two-moon' dataset experiment. To assess the effectiveness of VCReg, we replicated this setting with a three-layer network and applied our method during training. Our visualized results in Figure 1 make it apparent that VCReg has a marked advantage over traditional regularization techniques, particularly in the aspects of separation margins. Thus, it is reasonable to conclude that VCReg can help mitigate gradient starvation.

These results are significant for multiple reasons. Firstly, the encouraging outcomes with the 'two-moon' synthetic dataset set the stage for investigating VCReg's applicability in more complex, high-

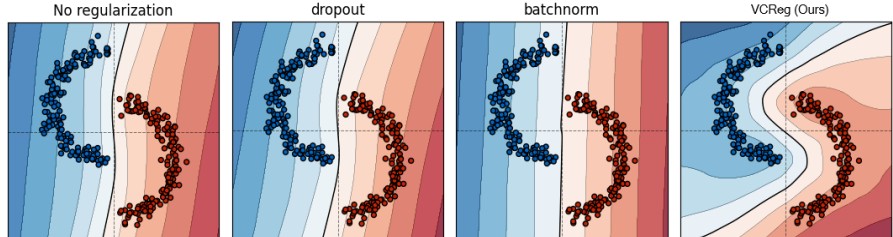

**Figure 1: Comparative Evaluation of VCReg and Traditional Regularization Techniques on a 'Two-Moon' Synthetic Dataset.** Decision boundaries are averaged over ten distinct runs with random data point sampling and model initialization. A single run's data points are displayed for visual clarity. The contrast between VCReg and conventional methods underscores the latter's limitations in forming intricate decision boundaries, while highlighting VCReg's effectiveness in generating meaningful ones.

dimensional tasks, thus cementing its status as a potent tool in contemporary machine learning. Second, VCReg's capability to mitigate gradient starvation indicates that neural networks trained using this method excel at learning complex, non-linear mappings—an essential trait for tasks that require a sophisticated understanding of data distributions. Lastly, VCReg surpasses traditional regularization techniques by generating a feature space that is both discriminative and rich in information. This highlights its potential to boost the generalizability of neural networks, which is crucial in real-world scenarios where models need to be both robust and flexible.

## 5.2 Preventing Neural Collapse and Information Compression

To deepen our understanding of VCReg and its training dynamics, we closely examine its learned representations. A recent study (Papyan et al., 2020) observed a peculiar trend in deep networks trained for classification tasks: The top-layer feature embeddings of training samples from the same class tend to cluster around their respective class means, which are as distant from each other as possible. However, this phenomenon could potentially result in a loss of diversity among the learned features (Papyan et al., 2020), thus curtailing the network's capacity to grasp the complexity of the data and leading to suboptimal performance (Li et al., 2018) for transfer learning.

Our investigation is based on two key metrics:

**Class-Distance Normalized Variance (CDNV)** For a feature map $f : \mathbb{R}^d \to \mathbb{R}^p$ and two unlabeled sets of samples $S_1, S_2 \subset \mathbb{R}^d$, the CDNV is defined as

$$V_f(S_1, S_2) = \frac{\text{Var}_f(S_1) + \text{Var}_f(S_2)}{2\|\mu_f(S_1) - \mu_f(S_2)\|^2}, \tag{8}$$

where $\mu_f(S)$ and $\text{Var}_f(S)$ signify the mean and variance of the set $\{f(x) \mid x \in S\}$. This metric measures the degree of clustering of the features extracted from $S_1$ and $S_2$, in relation to the distance between their respective features. A value approaching zero indicates perfect clustering.

**Nearest Class-Center Classifier (NCC)** This classifier is defined as

$$\hat{h}(x) = \arg\min_{c \in [C]} \|f(x) - \mu_f(S_c)\| \tag{9}$$

According to this measure, during training, collapsed feature embeddings in the penultimate layer become separable, and the classifier converges to the 'nearest class-center classifier'.

**Preventing Information Compression** We next address the prevention of information compression during the learning process. Although effective compression often yields superior representations, overly aggressive compression might cause the loss of crucial information about the target task (Shwartz-Ziv et al., 2018; Shwartz-Ziv & Alemi, 2020; Shwartz-Ziv & LeCun, 2023). To investigate this, we use the mutual information neural estimation (MINE) (Belghazi et al., 2018), a method specifically designed to estimate the mutual information between the input and its corresponding embedded representation. This metric effectively gauges the complexity level of the representation, essentially indicating how much information (in terms of number of bits) it encodes.

**Table 6: VCReg learns richer representation and prevents neural collapse and information compression** Metrics include Class-Distance Normalized Variance (CDNV), Nearest Class-Center Classifier (NCC), and Mutual Information (MI). Higher values in each metric for the VCReg model indicate reduced neural collapse and richer feature representations.

| Network | CDNV | NCC | MI |
|---|---|---|---|
| ConvNeXt | 0.28 | 0.99 | 2.8 |
| ConvNeXt (VCReg) | **0.56** | **0.81** | **4.6** |

We evaluate the learned representations of two ConvNeXt models (Liu et al., 2022), which are trained on ImageNet with supervised loss. One model was trained with VCReg, while the other was trained without VCReg. As demonstrated in Table 6, both types of collapse, measured by CDNV and NCC, and the mutual information estimation reveal that VCReg representations have significantly more diverse features (lower neural collapse) and contain more information compared to regular training. This suggests that not only does VCReg achieve superior results, but also its underlying representation contains more information.

In summary, the VCReg method mitigates the neural collapse phenomenon and prevents excessive information compression, two crucial factors that often limit the effectiveness of deep learning models in transfer learning tasks. Our findings highlight the potential of VCReg as a valuable addition to the deep learning toolbox, significantly increasing the generalizability of learned representations.

## 6 CONCLUSION

In this work, we addressed prevalent challenges in supervised pretraining for transfer learning by introducing Variance-Covariance Regularization (VCReg). Building on the regularization technique of the self-supervised VICReg method, VCReg is designed to cultivate robust and generalizable features. Unlike conventional methods that attach regularization only to the final layer, we strategically incorporate VCReg across intermediate layers to optimize its efficacy.

Our key contributions are threefold:

1. We present a computationally efficient VCReg implementation that is adaptable to various network architectures.

2. We provide empirical evidence through comprehensive evaluations on multiple benchmarks, demonstrating that using VCReg yields notable improvements in transfer learning performance across various network architectures and different learning paradigms.

3. Our in-depth analyses confirm VCReg's effectiveness in overcoming typical transfer learning hurdles such as neural collapse and gradient starvation.

To conclude, VCReg stands out as a potent and adaptable regularization technique that elevates the quality and applicability of learned representations. It enhances both the performance and reliability of models in transfer learning settings, and paves the way for further research aimed at achieving highly optimized and generalizable machine learning models.

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

## A  Experimental Investigation on Effective Application of VCReg to Standard Networks

To determine the optimal manner of integrating the VCReg into a standard network, we conducted several experiments utilizing the ConvNeXt-Atto architecture, trained on ImageNet following the torchvision (Paszke et al., 2019) training recipe. To reduce the training time, we limited the network training to 90 epochs with a batch size of 4096. The complete configuration comprised 90 epochs, a batch size of 4096, two learning rate of $\{0.016, 0.008\}$ with a 5 epochs linear warmup followed by a cosine annealing decay. The weight decay was set at $0.05$ and the norm layers were excluded from the weight decay. we experimented with $\alpha \in \{1.28, 0.64, 0.32, 0.16\}$ and $\beta \in \{0.16, 0.08, 0.04, 0.02, 0.01\}$.

We experimented with incorporating the VCReg layers in four different locations:

1. Applying the VCReg exclusively to the second last representation (the input of the classification layer).
2. Applying VCReg to the output of each ConvNeXt block.
3. Applying VCReg to the output of each downsample layer.
4. Applying VCReg to the output of both, each ConvNeXt block and each downsample layer.

The VCReg layer was implemented as detailed in 1, with the addition of a mean removal layer along the batch preceding the VCReg layer to ensure that the VCReg input exhibited a zero mean.

**Table 7: Transfer Learning Experiments with Different VCReg Configurations**

| Architecture | Food | Cars | Aircraft | Pets | Flowers | DTD |
|---|---|---|---|---|---|---|
| ConvNeXt-Atto (VCReg1) | 63.2% | 39.6% | 55.9% | 89.1% | 85.3% | 65.1% |
| ConvNeXt-Atto (VCReg2) | **66.8%** | 48.1% | **60.4%** | **91.1%** | **86.4%** | **66.4%** |
| ConvNeXt-Atto (VCReg3) | 64.0% | 40.9% | 56.5% | 89.4% | 85.9% | 65.1% |
| ConvNeXt-Atto (VCReg4) | 66.7% | **48.3%** | 59.6% | 90.6% | 85.6% | 66.1% |

The results in Table 7 indicate superior performance when the VCReg layer is applied to the output of each block (second setup) or applied to the output of blocks and downsample layers (fourth setup) compared to the other setups. Considering architectures like ViT lack downsample layers, for consistency across different architectures, we decided to use this configuration for further experiments.

## B  The Fast Implementation of the VCReg

The VCRegeg does not affect the forward pass in any way, allowing us to substantially speed up the implementation by modifying the backward function directly. Instead of computing the VCReg loss and backpropagating it, we can directly alter the calculated gradient. This is possible since the VCReg loss calculation only requires the current representation. The specifics of this speed-optimized implementation are outlined in Algorithm 1.

## C  Implementation Details

### C.1  Transfer Learning Experiments with ImageNet Pretraining

In conducting the transfer learning experiments, we adhered primarily to the training recipe specified by PyTorch Paszke et al. (2019) for each respective architecture during the supervised pretraining phase. We abstained from pretraining any of the baseline models, instead opting to directly download the weights from PyTorch's own repository. The only modifications applied were to the parameters associated with VCReg loss, and we experimented with $\alpha \in \{1.28, 0.64, 0.32, 0.16\}$ and $\beta \in \{0.16, 0.08, 0.04, 0.02, 0.01\}$.

For iNaturalist 18 Van Horn et al. (2018) and Place205 Zhou et al. (2014), we relied on the experimental settings detailed in Zbontar et al. (2021) for the linear probe evaluation.

---

**Algorithm 1:** PyTorch-Style Pseudocode for Fast VCReg Implementation

---

```
# α, β and ε :  hyperparameters
# mm:  matrix-matrix multiplication

class VarianceCovarianceRegularizationFunction(Function):
    # forward pass
    # We assume the input has zero mean per channel
    # In practice, we apply a batch demean operation before call the function
    def forward(ctx, input):
        ctx.save_for_backward(input)
        return input
    # backward pass
    def backward(ctx, grad_output):
        input, = ctx.saved_tensors
        # reshape the input to have (n, d) shape
        flattened_input = input.flatten(start_dim=0, end_dim=-2)
        n, d = flattened_input.shape
        # calculate the covariance matrix
        covariance_matrix = mm(flattened_input.t(), flattened_input) / (n - 1)
        # calculate the gradient
        diagonal = F.threshold(rsqrt(covariance_matrix.diagonal() + \epsilon), 1.0, 0.0)
        std_grad_input = diagonal * flattened_input
        cov_grad_input = torch.mm(flattened_input, covariance_matrix.fill_diagonal_(0))

        grad_input = grad_output
                    - α/(d(n-1)) * std_grad_input.view(grad_output)
                    + 4β/(d(d-1)) * cov_grad_input

        return grad_input
```

---

Regarding Food-101 Bossard et al. (2014), Stanford Cars Krause et al. (2013), FGVC Aircraft Maji et al. (2013), Oxford-IIIT Pets Parkhi et al. (2012), Oxford 102 Flowers Nilsback & Zisserman (2008), and the Describable Textures Dataset (DTD) Cimpoi et al. (2014), we complied with the evaluation protocol provided by Chen et al. (2020); Kornblith et al. (2021). An $L2$-regularized multinomial logistic regression classifier was trained on features extracted from the frozen pretrained network. Optimization of the softmax cross-entropy objective was conducted using L-BFGS, without the application of data augmentation. All images were resized to 224 pixels along the shorter side through bicubic resampling, followed by a 224 x 224 center crop. The $L2$-regularization parameter was selected from a range of 45 logarithmically spaced values between 0.00001 and 100000.

All experiments were run three times, with the average results presented in Table 2.

## C.2    Subclass Linear Probing Result with Network Pretrained on Superclass Label

For our subclass linear probing experiments, we employed a ConvNeXt-Atto network. Each model was pretrained for 200 epochs using the superclasses, adhering to the same procedure detailed in the Appendix A. Subsequent to this pretraining phase, we initiated a linear probing process using the subclass labels. This linear classifier was trained for 100 epochs, using a base learning rate of 0.016 in conjunction with a cosine learning rate schedule. The optimizer used was AdamW, which worked to minimize cross-entropy loss with a weight decay set at 0.05. We processed our training data in batches of 256.

## C.3    Long-Tail Learning Result

For our long-tail learning experiments, we use ResNet-32 as a backbone for experiments on the CIFAR10-LT and CIFAR100-LT datasets. We trained 100 epochs with batch size 256, Adam optimizer with two learning rate of $\{0.016, 0.008\}$ with a 10-epoch linear warm-up followed by a cosine annealing decay. The weight decay was set at 0.05 and the norm layers were excluded from the weight decay. we experimented with $\alpha \in \{1.28, 0.64, 0.32, 0.16\}$ and $\beta \in \{0.16, 0.08, 0.04, 0.02, 0.01\}$.

## C.4 VCReg with Self-Supervised Learning Methods

We trained a ResNet-50 model in four different setups, using either the SimCLR loss or the VICReg loss with the ImageNet dataset. The application of the VCReg is the same as described in Appendix A.

We closely follow the original setting in Chen et al. (2020) for SimCLR pretraining and Bardes et al. (2021) for VICReg pretraining.

**Augmentation** - For both methods, we use the same augmentation methods. Each augmented view is generated from a random set of augmentations of the same input image. We apply a series of standard augmentations for each view, including random cropping, resizing to 224x224, random horizontal flipping, random color-jittering, randomly converting to grayscale, and a random Gaussian blur. These augmentations are applied symmetrically on two branches Geiping et al. (2022)

**Architecture** - For SimCLR, the encoder is a ResNet-50 network without the final classification layer followed by a projector. The projector is a two-layer MLP with input dimension 2048, hidden dimension 2048, and output dimension 256. The projector has ReLU between the two layers and batch normalization after every layer. This 256-dimensional embedding is fed to the infoNCE loss.

For VICReg, the online encoder is a ResNet-50 network without the final classification layer. The online projector is a two-layer MLP with input dimension 2048, hidden dimension 8192, and output dimension 8192. The projector has ReLU between the two layers and batch normalization after every layer. This 8192-dimensional embedding is fed to the infoNCE loss.

For VCReg, we just applied the VCReg layers to the ResNet-50 network as described in the Appendix A.

**Optimization** - We follow the training protocol in Zbontar et al. (2021). For SimCLR experiments, we used a LARS optimizer and a base learning rate 0.3 with cosine learning rate decay schedule. We pretrain the model for 100 epochs with 5 epochs warm-up with batch size 4096.

For VICReg, we use a LARS optimizer and a base learning rate 0.2 using cosine learning rate decay schedule. We pretrain the model for 100 epochs with 5 epochs warm-up with batch size 4096.

**Evaluation** we followed the standard evaluation protocol as prescribed by Misra & Maaten (2020); Zbontar et al. (2021), performing linear probing evaluations, on iNaturalist 18 Van Horn et al. (2018) and Place205 Zhou et al. (2014) datasets.

