# OpenReview forum: "Variance-Covariance Regularization Improves Representation Learning"
_ICLR.cc/2024/Conference — ICLR 2024 Conference Withdrawn Submission_

### Official Review · Reviewer_NHSr · 2023-10-13

**Soundness:** 1 poor
**Presentation:** 1 poor
**Contribution:** 1 poor
**Rating:** 3
**Confidence:** 4

**Summary:**

This paper proposes a self-supervised learning regularization techniques adapted from VCReg method to bolster feature transferability. Generally, the wirting of paper is not good, such as lack of definition of notations and some subjective claimations, the problem is not well stated, the method is not convincing and seems unreasonable.

**Strengths:**

- __[Good empirical results]__ The empirical results are good.

**Weaknesses:**

__Main concerns:__
- __[Lack of novelty]__ The proposed method merely simply applies the regularization proposed in VICReg on intermediate layers;
- __[Unclear Motivation]__ The motivation of this paper is not well illustrated;
- __[Poor writing]__ The writing of the paper is not good, it lacks preliminary knowledge and is a little hard to read.
- __[Lack of statements of problem]__ The paper claims that pre-training phase undermines feature transferability and feature transferability can be specifically expressed as gradient starvation and neural collapse. However, it is difficult to convince the connections. See Q2.
- __[Unconvincing experiments]__ See Question 4.


__Minor concerns:__
- __[Lack of illustration of motivation]__ The description/introduction in Intro section fails to provide a clear explanation of how the method works/how the VCReg is constructed and the basic/intuitive principles regarding the method.
- __[Lack of preliminary knowledge]__ Although invariance loss $\ell_{\rm inv}$ in Eq. (1) is mentioned above, an explanation for this notation is necessary. Alternatively, some preliminary knowledge ought to be presented before the concrete introduction of the proposed method, such as the concrete formulation of invariance criterion $\ell_{\rm inv}$, to make the full paper more complete. Similarly, for readers who are familiar with VICReg, invariant criterion is a mean-square distance function. However, for those who are not familiar with, it is a little bit hard to understand what invariant criterion is.

**Questions:**

1. The authors claim that pre-training phase undermines feature transferability by prioritizing features that minimize the pre-training loss. Although some references are provided, it is still not very intuitive to readers. More simple yet typical illustrations will facilitate to convince the motivation of this paper. In addition, some concrete explanation of why pre-training tends to select such features will also help.
2. At the beginning of paper, authors claim the poor feature transferability problem resulted from exsiting in pre-training phase since the model tends to prioritize the features that minimize the loss. Then discuss that some concrete cases regarding this, such as gradient starvation and neural collapse problem. What are the relationships between the aforementioned problem and latter two problems? Why they result in prioritizing features that minimize the loss? Does the phenomenon in SSL that output identical and constant featrues belong to this case? If so, why you think that it is the model 'focus on' some specific features and what are the characteristics of these features? I think that such statement is far-fetched and requires more careful definition.
3. As mentioned in Related Works section, variance term helps learn a wide range of features, but why it intutively learns the unique, non-redundant information (Page 4)?
4. I do not think that experimental results in Appendix A are convincing. My concerns mainly include three aspects:
    - (1) First of all, when applying VCReg to all blocks, what does the training loss look like? Are all VCReg added or averaged?
    - (2) Secondly, why different $\alpha$ and $\beta$ is set? Why not use the same $\alpha$ and $\beta$ for fairness?
    - (3) Since the training time is limited to 90 epochs, it exists the case that the model does not converge. For example, Flowers dataset is simple datasets and may converge fast while DTD dataset is difficult and requires more time for model to converge. If results of converged model and not converaged model are compared, it is unfair.
5. Due to Question 4, I further don't believe that the motivation of applying VCReg to intermediate layers is strong and convincing.
6. What are the evidence/references or empirical results for the statement "Regularizing intermediate layers enable the model to capture more complex, higher-level ...... model's generalization capabilities. Furthermore, it fosters the development of feature hierarchies ...... improved transfer learning performance"? Four concerns are included here:
    - The statement is not intuitive.
    - Besides, in general, it is widely believed that intermediate layers produce relatively low level features compared with the final output. Why do you think that regularization contributes to learn complex and high-level features?
    - Why minimizing covariance helps improve the stability and generalization performance. Could you please provide some empirical results, such as learning curves, and results or references about generalization bound improvements?
    - From my understanding, maximizing the variance aims to help network avoid identical and constant output which further helps avoid model collapse, why it can learn hierarchies and enrich latent space?
7. What are the concrete details of operation metioned in "Spatial Dimension"? Could you express it in mathematical way?
8. I am also confused with reasons and priciples behind ${\rm SmoothL1(x)}$? What are the insights behind Eq. (7): Will the modification operation be involved in the optimization process? How does such operation affect the optimization process.
9. I do not understand the fast implementation of VCReg in Section 3.3. Since the regularizations are calculated on the corresponding intermediate representations, when calculating gradients, they will not affect the subsequent modules. However, the modules in front of them will be affected due to the chain rules. Since gradients in this paper is calculated manually, will the gradients be modified? In fact, I can not get your operations in Algorithm 1 since this paper does not provided a systematic mathematical formulation. I am not sure Algorithm 1 is correct.
10. Since the gradients are manually set, is the ${\rm SmoothL1}$ operation taken into consideration?

The most important question is:  __how does the proposed method solve the problem that the pretraining phase undermines feature transferability?__ Since this paper fails to formulate this problem. I do not think this paper is sound.

__With all concerns and questions mentioned above take into consideration, I do not check the experiments part because I am not sure I can evaluate the results correctly due to my concerns and questions mentioned above. If my concerns and questions are well addressed later, I will further check the experiments part during rebuttal phase.__

__Some minor problems:__
- What is the definition of $\ell_{\rm sup}(\cdot, \cdot)$?

---

### Official Review · Reviewer_mHKo · 2023-10-26

**Soundness:** 2 fair
**Presentation:** 3 good
**Contribution:** 2 fair
**Rating:** 5
**Confidence:** 4

**Summary:**

A regularization (variance, covariance) from self-supervised contrastive learning (SSL) (VICReg paper) is applied to supervised learning (SL) showing improvements in downstream tasks, long-tail learning, and hierarchical classification.

**Strengths:**

+ Inspired by the success in SSL applying to SL may be interesting.
+ Writing is clear
+ Some improvements are verified when performing supervised pretraining on ImageNet and transferring to another dataset.

**Weaknesses:**

+ I am concerned about the novelty because VICReg regularization can support the other SSLs such as SimCLR/BYOL/SimSiam/MoCo/... since its objective loss (works on the dimension elements along the batch) does not conflict with the contrastive loss, I think its helpfulness to SSL or SL is not surprising. Paper [1] already verified that using regularization similar to covariance in Barlow Twins (without invariance) could already benefit many learning methods including SSLs and SL. Even DimCL [1] only uses (similar) covariance regularization (without invariance) for the final layer (projector) already yielded satisfactory performance and does not need to apply to many layers as the proposed method which would bring a huge computation and make it overcomplicated.

+ Comparing the downstream tasks of only supervised learning methods (Table 2) might not be very meaningful because the drawback of supervised learning requires massive manual labels (ImageNet) which is prohibitively computational cost. Many SSL methods have already shown that can significantly outperform SL on downstream tasks including both classification and object detection. This makes the contribution of this paper less impactful.

[1] DimCL: Dimensional Contrastive Learning for Improving Self-Supervised Learning, IEEE Access 2023.

**Questions:**

See weaknesses

---

### Official Review · Reviewer_qZNE · 2023-10-27

**Soundness:** 2 fair
**Presentation:** 4 excellent
**Contribution:** 2 fair
**Rating:** 3
**Confidence:** 4

**Summary:**

The paper adapts a self-supervised learning regularization technique from VICReg to the supervised learning contexts. The proposed method encourages the network to learn high-variance and low-covariance representations to promote the diversity of features. The method is termed as VCReg. To validate the effectiveness of VCReg, the paper evaluates it in various settings of image classification tasks such as transfer learning, long-tail learning, and hierarchical classification. Further, the paper analyzed the gradient starvation and neural collapse problems to explore why VCReg helps the transfer learning task.

**Strengths:**

1. The proposed method is easy to implement and compatible with different neural network architectures.

2. The paper provides experiment results on different downstream tasks such as transfer learning, long-tailed, and hierarchical classification. The further exploration of VCReg on gradient starvation and neural collapse provides some fresh insights and directions on what types of feature representations can facilitate transfer learning.

3. The presentation of this paper is well-organized and clear.

**Weaknesses:**

1. The paper mainly addresses the benefits of VCReg on transfer learning. However, I think experiments about transfer learning cannot fully support the conclusion since experiments on evaluating the generalization ability are not provided. For example, I think to specifically validate the transfer learning ability, domain generalization [1,2] experiments should be provided.

2. Similarly, the experiment of long-tailed tasks should also be evaluated on the latest benchmarks and compared with the latest methods [3,4].

3. The overall improvement in self-supervised learning is incremental. On some datasets in Tables 4 and 2, the performance decreases. Also, in Table 4 ImageNet results of SimCLR, there may be a typo (67.2% vs 67.1%). 67.2% should be in bold.

[1]. https://wilds.stanford.edu/datasets/

[2]. https://github.com/facebookresearch/DomainBed

[3]. Long-Tailed Recognition Using Class-Balanced Experts

[4]. ImageNet-LT

**Questions:**

1. The relation between gradient starvation, neural collapse, and transfer learning is not very clear. Could the author elaborate more on how VCReg mitigates these issues and further improves the generalization ability?

2. For Table 4, when the author applied VCReg, does the method include batch normalization?

3. Besides, in section 3.2 line 4, the author mentions that the 'strategy can minimize the internal covariate shift across layers, which in turn improves both the stability of training and the model’s generalization
capabilities'. I am wondering if the author could elaborate more on this. Is there any experimental evidence to support that VCReg can mitigate the internal covariate shift and further benefit the training? Since there is research arguing that BN does not reduce the internal covariate shift [1], I am interested in whether the VCReg can reduce it, and improve the training.

3. I am also wondering about the compatibility of VCReg with other types of SSL methods such as BYOL.

[1]. How Does Batch Normalization Help Optimization?

---

### Official Review · Reviewer_DtqU · 2023-10-29

**Soundness:** 2 fair
**Presentation:** 3 good
**Contribution:** 2 fair
**Rating:** 5
**Confidence:** 3

**Summary:**

The paper proposes to use variance-covariance regularization (VCReg) to improve transfer learning performance of models trained either using supervised learning (SL) and self-supervised learning (SSL). The regularizer is inspired by VICReg which is a very popular and strong baseline in SSL. A key aspect of VCReg is to use regularizers on various layer outputs and not just the final layer. The paper conducts an empirical analysis using ImageNet-1K as a source domain dataset to test the effectiveness of VCReg. The paper also hints at VCReg's ability to combat gradient starvation and neural collapse that in turn make VCReg effective at transfer learning.

**Strengths:**

- The paper argues for improving transfer learning which may be important in this era of large model pretraining and using features from large models for downstream work. The motivation is sound
- The idea of using variance and covariance terms in supervised learning is very interesting and not something that the reviewer has seen before in supervised learning
- Empirical validation is extensive in terms of the datasets considered with ImageNet-1K being considered the source dataset. The architectures considered are commonly used in practice so there is good coverage on this component as well.
- Results generally support the hypothesis that VCReg when carefully designed and applied can work well over not using it

**Weaknesses:**

- The method, VCReg, while being generically described is missing several critical details in the paper. The idea of using VCReg on several layers instead of just the layer that outputs representation is not probably described in terms of how the layers are chosen and whether the weights were varied on a per-layer basis. A proper ablation would be very useful for both the readers and users of this method that want to test the approach in their implementations.

- The authors may want to describe gradients derivation in the main paper and include full derivations of the gradients for VCRegin the appendix.  This information is key in making their method work in practice

- The connection to gradient starvation is very weak at best. The paper only shows an experiment with a toy dataset that is meant to build intuition on how various methods avoid gradient starvation. On a related note, have the authors considered running spectral decoupling introduced in gradient starvation paper and compare to VCReg?

- The connection to neural collapse is interesting but the empirical evidence is limited (understandably so due to compute but I feel its my duty to call this out here)

- I am curious about how VCReg was applied in the SSL setting? Would VCReg essentially be VICReg with internal layers getting regularized with VCReg trick? Would using VICReg for intermediate layers make sense here?

**Questions:**

- Please check the comments and questions outlined in **Weaknesses** section

- Wightman et. al. in "ResNet Strikes Back" show that one can achieve better accuracy with ImageNet-1K using an improved training strategy? Have the authors considered using this approach as a baseline? If so, do they see a similar improvement to what is shown in the paper?

- Please consider moving some of the writing on key aspects of the paper from the appendix to main paper

- I did not quite understand what the paper means in "Spatial Dimensions" paragraph. Could the authors please update their explanation and/or include more details in the appendix?